# Helper Innate Lymphoid Cells—Unappreciated Players in Melanoma Therapy

**DOI:** 10.3390/cancers15030933

**Published:** 2023-02-01

**Authors:** Cinzia Garofalo, Annamaria Cerantonio, Carolina Muscoli, Vincenzo Mollace, Giuseppe Viglietto, Carmela De Marco, Costanza Maria Cristiani

**Affiliations:** 1Department of Experimental and Clinical Medicine, “Magna Græcia” University of Catanzaro, 88100 Catanzaro, Italy; 2Department of Health Science, Institute of Research for Food Safety & Health (IRC-FSH), “Magna Græcia” University of Catanzaro, 88100 Catanzaro, Italy

**Keywords:** innate lymphoid cells, melanoma, tumor microenvironment, immune therapy, immune checkpoints inhibitors, targeted therapy

## Abstract

**Simple Summary:**

Helper innate lymphoid cells (ILCs) represent the innate counterpart of helper T lymphocytes. Increasing findings indicate that they are involved in the pathogenesis and progression of solid tumors. In this review, we describe the alterations induced by the melanoma tumor microenvironment (TME) in helper ILCs, as well as their capability to respond to melanoma cells and shape the TME in turn. Next, we provide details about the expression and function of checkpoint receptors on helper ILCs and how therapy with immune checkpoint inhibitors may modulate their activation and properties. We also describe how the immunomodulatory properties of targeted therapy used to treat melanoma may affect helper ILC function. We conclude by discussing the limits of the current knowledge about helper ILC roles in metastatic melanoma disease, as well as the challenges that need to be addressed to clarify their contribution to pathophysiology and therapy.

**Abstract:**

Immune checkpoint inhibitors (ICIs) and targeted therapy have dramatically changed the outcome of metastatic melanoma patients. Although immune checkpoints were developed based on the biology of adaptive T cells, they have subsequently been shown to be expressed by other subsets of immune cells. Similarly, the immunomodulatory properties of targeted therapy have been studied primarily with respect to T lymphocytes, but other subsets of immune cells could be affected. Innate lymphoid cells (ILCs) are considered the innate counterpart of T lymphocytes and include cytotoxic natural killer cells, as well as three helper subsets, ILC1, ILC2 and ILC3. Thanks to their tissue distribution and their ability to respond rapidly to environmental stimuli, ILCs play a central role in shaping immunity. While the role of NK cells in melanoma physiopathology and therapy is well established, little is known about the other helper ILC subsets. In this review, we summarize recent findings on the ability of the melanoma TME to influence the phenotype and functional plasticity of helper ILCs and highlight how this subset may in turn shape the TME. We also discuss changes in the melanoma TME induced by targeted therapy that could affect helper ILC functions, the expression of immune checkpoints on this subset and how their inhibition by ICIs may modulate helper ILC function and contribute to therapeutic efficacy.

## 1. Introduction

Among skin tumors, malignant melanoma is characterized by the highest rate of metastasization and resistance to conventional therapies, making it the most lethal type of cutaneous cancer. Indeed, disease outcomes following standard chemotherapies have been less than satisfactory, with 5-year survival rates of approximately 10% [1]. However, the landscape of melanoma treatment has been dramatically changed from 2011 thanks to the introduction of two distinct classes of therapeutics: targeted therapies and immunotherapy with immune checkpoints inhibitors (ICIs).

As the name suggests, targeted therapies were designed to target cell-intrinsic pathways aberrantly activated in melanoma cells. The most commonly altered signaling pathway in metastatic melanoma is the BRAF/MAPK/ERK pathway. About 50% of mutations occur at codon 600 of the *BRAF* gene, causing the substitution of a valine (V), usually with glutamic acid (E), and the constitutive activation of the kinase and thus the entire pathway [2,3]. Indeed, the first targeted drugs approved for metastatic melanoma (vemurafenib, dabrafenib and encorafenib) specifically target the *BRAFV600E* mutation [4,5,6]. Moreover, since MAPK/ERK reactivation was commonly observed as a resistance mechanism, further MEK inhibitors (binimetinib, cobimetinib and trametinib) have been developed and used in combination with BRAF inhibitors [7,8,9].

On the other hand, ICIs have been designed to target the inhibitory receptors physiologically limiting over-activation of the immune system but of which expression is often dysregulated in cancer [10]. Since 2011, ICIs targeting the cytotoxic T-lymphocyte antigen 4 (CTLA-4) and programmed cell death protein 1 (PD-1) have been approved for the treatment of metastatic melanoma and are currently used in clinical practice [11,12,13,14]. Moreover, other ICIs are being developed and tested to treat those patients that do not respond to current immunotherapies [15,16].

Since the two classes of drugs act by different mechanisms, their efficacy profiles are significantly different. With BRAF and MEK inhibitors, a remarkable clinical response can be seen in most treated patients. However, the therapy can only be administrated to mutation-harboring patients, and the response is usually short-lived. On the contrary, ICIs act slowly, and only a fraction of patients responds to the treatment. Nevertheless, they can be administered regardless of the mutational status, and the achieved response is usually durable [17]. This complementarity opened the possibility to combine the two kinds of drugs, further supported by the evidence that targeted therapy also possesses immunomodulatory properties [18].

Since T lymphocytes have long been considered the most important players in anti-cancer immunity, they have not only been the primary target for the development of ICIs but also the major immune cell subset investigated as a biomarker for ICI effectiveness [19]. Similarly, the capability of targeted therapy to positively modulate the tumor microenvironment (TME) has been primarily studied with respect to them [17,18]. However, T cells are not the only component of immune cells involved in cancer immunity. In particular, innate lymphoid cells (ILCs), which are considered the innate counterpart of T lymphocytes [20], have been shown to infiltrate tumors and to express inhibitory receptors [21,22,23,24,25,26,27]. This suggests that ILCs may also be affected by both ICIs and targeted therapies, contributing to their clinical effect. However, the role of ILCs in tumor immunology is still controversial and not fully understood.

Previous reports in this field have described ILCs in the general context of cancer, without focusing on a specific pathological entity [21]. Here, we specifically discuss the role of ILCs in the pathophysiology and therapy of melanoma. Moreover, in most reviews on ILCs, a substantial part of the discussion is devoted to cytotoxic NK cells, which represent the best characterized ILC subpopulation in cancer [21,22,23]. Therefore, we have chosen to focus our attention specifically on other helper subpopulations, usually described to a minor extent.

In this review, we provide an overview of the evidence supporting an active role of helper ILCs in the context of melanoma pathogenesis and response to ICIs and targeted therapy, highlighting their potential contribution to clinical effectiveness. Specifically, we focus on (i) the direct cross-talk between helper ILCs and melanoma cells, (ii) the ability of helper ILCs to shape the melanoma TME, (iii) the expression of current targets of ICI therapy on helper ILCs, and (iv) the immunological impact of targeted therapy on helper ILC function.

## 2. Helper Innate Lymphoid Cells in Melanoma

Innate lymphoid cells (ILCs) are a heterogenous group of cells characterized by three shared features: lymphoid morphology, absence of rearranged antigen-specific receptors (BCR and TCR) and lack of myeloid markers [28]. Currently, ILCs are divided in five subsets: natural killer (NK) cells, lymphoid-tissue inducer (LTi) cells and helper ILCs, further distinguished as ILC1, ILC2 and ILC3 subsets [20]. Based on their effector functions and expression of master transcription factors, ILCs are considered the innate counterpart of T cells [29]. Similar to CD8+ T cells, NK cells are cytotoxic lymphocytes equipped with perforin and granzymes and able to release interferon γ (IFN-γ) and tumor necrosis factor α (TNFα) in order to kill target cells. Additionally, they require the same transcription factors, T-bet and Eomes, for their development. ILC1s, ILC2s and ILC3s parallel T helper (Th)1, Th2 and Th17 cells, respectively. ILC1s express T-bet and produce both IFN-γ and TNFα; ILC2s depend on GATA3 for their development and secrete type 2 cytokines, such as interleukin (IL)-4, IL-5 and IL-13; ILC3s recapitulate the Th17 phenotype by secreting IL-17 and IL-22 and expressing RORγt. Lastly, LTi cells, similar to ILC3s, are involved in the development of secondary lymphoid tissues during embryogenesis [20].

As members of innate immunity, ILCs represent one of the first barriers against pathogens and malignant cells. Indeed, ILCs are able to promptly react to a wide range of stimuli, including danger signals, cytokines and stress-associated molecules expressed by tissue cells. In response, ILCs produce several different cytokines, thus playing a central role in shaping subsequent innate and adaptive immunity. Moreover, ILCs are highly plastic, which means that they can switch from a phenotype to another based on environmental signals, finely tuning immune responses [29]. Therefore, ILCs might be important in tumorigenesis and the net effect of ILCs on the anti-tumoral immune response will depend on the recruited subpopulation, the produced cytokines and signals delivered by the TME. Of notice, all of these mechanisms might be modulated by ICIs.

The anti-tumor properties of NK cells in the context of melanoma are unquestionable and have been already reviewed by us elsewhere [22,23]. On the other hand, the role of the other ILC subsets has only recently been investigated and is still debated. Thus, in the next paragraphs, we will focus the discussion on helper ILCs.

### 2.1. ILC1s in Melanoma

Human ILC1s represent an elusive and probably heterogenous population since, except for the a-chain of the IL-7 receptor CD127, they do not express specific lineage markers and partly overlap phenotypically with NK cells [30]. Expression levels of CD127 distinguishes ILC1s into two main subsets: CD127low ILC1s share phenotypical and functional similarities with NK cells since they respond to IL-15 and IL-12 and express prototypical NK cell markers, such as CD56, CD94 and NKp44, as well as perforin and Eomes [31,32]; on the other hand, CD127high ILC1s lack the expression of these surface molecules and are responsive to IL-12 and IL-18 [32,33]. Therefore, ILC1s are usually considered the main helper ILC subset exerting anti-tumor activities, particularly in Th1 cytokine-enriched environments [29].

ILC1s have been shown to be expanded in peripheral blood and infiltrated lymph nodes of melanoma patients, as well as within the tumors of melanoma-bearing mice [24,34]. Moreover, the specific enrichment of CD56^+^CD94^+^ NK-like cells was observed within peripheral ILC1s [24]. However, the molecular mechanism driving this expansion, as well as its net effect on melanoma disease, is not known. It has been demonstrated that both ILC3s and ILC2s can transdifferentiate into ILC1s based on the environmental cues. Upon IL-12 stimulation, ILC3s down-regulate the expression of RORγt and shift toward ILC1s, as indicated by IFNγ secretion and T-bet up-regulation [32,33,35]. Since circulating ILC3s were concomitantly reduced in melanoma patients [24], expanded ILC1s might be derived from the peripheral conversion of ILC3s into ILC1s driven by IL-12, of which serum levels have been found to be increased in melanoma patients [36]. Alternatively, the altered frequencies of ILC1s and ILC3s could reflect transdifferentiation phenomena occurring within melanoma lesions due to IL-12 secretion by dendritic cells (DCs) [32]. Similarly, IL-12 also induces an ILC2 shift towards ILC1s [37,38,39]. However, alterations in peripheral ILC2 frequencies have not been reported in melanoma patients [24,34,40].

An alternative, intriguing possibility raised from murine models is that ILC1s would be derived from NK cells under the effect of TGF-β, which is increased in the TME and circulation of melanoma patients [23,41]. In these models, TGF-β signaling has been shown to promote the conversion of NK cells into NK-like ILC1s, characterized by an intermediate phenotype and high expression of inhibitory receptors [42,43]. This conversion appeared to be dependent on the non-canonical TGF-β signaling pathway in NK cells [43] (Figure 1A). A similar switch of NK cells towards an ILC1 phenotype under the influence of TGF-β has also been demonstrated in a human setting. In this context, CD56bright NK cells were the subset most prone to conversion into ILC1s, and the process was further enhanced by IL-15 [44]. Although differences in total peripheral NK cell frequency have not been reported in melanoma patients, CD56bright NK cells has been found to be expanded and to correlate with a worse outcome [22], which suggests the progressive conversion of cytotoxic CD56dim NK cells to the regulatory subset and then into ILC1s.

In addition, Ercolano et al. suggested that ILC1 enrichment in PBMCs from melanoma patients might be induced by melanoma cells through kynurenines, but they did not clarify whether it was due to ILC1 expansion or transdifferentiation of other subsets [34].

In contrast with the general view of ILC1s as protective, several reports actually indicated a pro-tumoral role for this subset in melanoma. Functional data from mouse models indicated that ILC1s derived from NK cells were not only unable to counteract tumor growth and metastasization but that they even promoted it, possibly because of the increased secretion of TNF-α and pro-angiogenic factors together with the poor production of IFN-γ and CCL5, which recruit and activate T and NK cells [42,43] (Figure 1A). Data obtained in human settings are in line with those from murine models, showing the reduced secretion of IFN-γ and the enhanced production of TNFα by ILC1s when they were co-cultured with melanoma cells [23,34]. The molecular mechanism by which ILC1s can interact with melanoma cells is not clear. Both murine and human ILC1s have been demonstrated to be equipped with surface receptors shared by NK cells, which recognize cognate ligands expressed by melanoma cells and affect NK cell-mediated killing [22,23,45]. However, the surface receptors involved in the cross-talk between ILC1s and melanoma cells, as well as the functional effect of such interactions, are still not known.

### 2.2. ILC2s in Melanoma

ILC2s probably represent the best defined human ILC subset due to the specific expression of the prostaglandin D2 receptor (CRTH2) [28]. In addition, ILC2s can express the stemness marker c-kit (CD117), which distinguishes ILC2s into two subpopulations: CD117^+^ ILC2s are more plastic and share some features with ILC3s, while CD117^-^ ILC2s are more committed towards a Th2-associated function [46,47]. Th2 responses have been usually associated with a tumor-supportive microenvironment due to the capability of Th2 cytokines to promote M2 polarization, the accumulation of myeloid-derived suppressor cells (MDSCs), Th2 lymphocytes and regulatory T cells (Tregs) and tissue repair. Therefore, ILC2s have been classically regarded as a pro-tumoral subset, and their abundance and/or activity have been found to be increased in several tumors [48]. However, in the melanoma setting, experimental evidence mainly points toward an anti-tumoral role for this subset.

Although major alterations in the peripheral frequency of ILC2s in melanoma patients have not been reported [24,34,40], this subset showed a low abundance within melanoma lesions in both human and murine settings [49,50]. As previously mentioned, this could be due to the transdifferentiation of ILC2s into ILC1s and/or ILC3s following melanoma TME stimuli [37,46,47]. Alternatively, ILC2 paucity could be due to TME acidification by melanoma cells, which has been shown to impair the survival and proliferation, as well as tumor infiltration, of this subset [49].

Manipulation of ILC2s in murine melanoma models showed that this subset affected melanoma progression, which was worsened when ILC2s were depleted and improved when they were stimulated, overall suggesting that ILC2s counteract melanoma development and progression. Different mechanisms have been elucidated explaining such an effect, mainly involving the activation and recruitment of eosinophils. In 2012, Ikutani and coworkers reported that, in a mouse model of melanoma lung metastasis, ILC2s responded to melanoma cells by secreting IL-5, which in turn promoted eosinophil recruitment and activation, thus counteracting metastasization [51] (Figure 1B). Similar findings were later confirmed by Wagner et al., who however did not further investigate the underlying molecular mechanism [49]. An analogous capability of ILC2s to restrain melanoma growth via eosinophil expansion and activation has also been described by Jacquelot et al. in a model of primary melanoma. However, in this context, the phenomenon was due to granulocyte–macrophage colony-stimulating factor (GM-CSF) secretion by ILC2s [50] (Figure 1B). In addition, two other mechanisms have been reported for ILC2s to directly restrain melanoma by inducing apoptosis thorough the secretion of the chemokines CXCL1 and CXCL2 in primary tumors and TNFα in lung metastases [52,53], suggesting that different anti-tumoral mechanisms may be specifically induced base on the TME. In all these models, ILC2 activation was mediated by the exogenous administration of IL-33. However, it is unclear whether the intra-tumoral levels of IL-33 are actually able to activate ILC2s. Moreover, human ILC2s have been shown to be able to respond to melanoma cells in vitro by up-regulating TNFα and IL-13 [24]; thus, it is possible that cell-to-cell interactions may contribute to activation [45].

On the other hand, a pro-tumoral role for IL-33-activated ILC2s in the context of melanoma has also been described. Particularly, two distinct murine models demonstrated that IL-33-stimulated ILC2s were able to negatively affect NK cell anti-melanoma activity and tumor rejection [54,55]. Again, different pathways have been implied, which could occur concomitantly. Following IL-33 stimulation, ILC2s have been shown to up-regulate the ectoenzyme CD73 [54], of which the product adenosine is well known to be enriched within the melanoma TME and to suppress NK cell cytotoxicity [29]. Moreover, ILC2s could suppress NK cells indirectly by recruiting eosinophils, which in turn limit glucose availability, needed for NK cell functions [55].

These conflicting effects of ILC2s on melanoma observed in vivo may be due to the different protocols used to stimulate ILC2s with IL-33, largely varying in terms of the amount, way and scheduling of administration. Thus, it is possible that the same cytokine, at different concentrations, may activate distinct pathways in ILC2s with divergent effects on melanoma development and progression.

### 2.3. ILC3s in Melanoma

ILC3 sare identified based on the expression of the stemness marker c-kit (CD117) and the natural cytotoxicity receptors (NCRs) NKp46 (in mice) and NKp44 (in humans). Particularly, NCRs functionally distinguish ILC3s into two subsets: NCR^-^ ILC3s secrete IL-17, whereas the production of IL-22 is restricted to NCR^+^ ILC3s [20]. However, recent evidence suggests that ILC3s may be a heterogeneous population containing ILC precursors, as well as more mature ILC3s [56]. Given the phenotypical and functional heterogeneity of this subset, as well as its involvement in both inflammation and tissue healing [57], the ILC3 role in melanoma is controversial.

In the periphery of melanoma patients, ILC3s have been shown to be reduced, particularly in the NCR^+^ component [24]. As previously mentioned, the total contraction of ILC3s could be due to their conversion into ILC1s under the influence of IL-12 [32,33,35], while TGF-β might promote the down-regulation of NCRs [58]. Thus, melanoma disease could trigger a complex pathway in which mature ILC3s regress into precursors under the influence of various cytokines and are then converted into ILC1s. Alternatively, ILC3s could directly switch into NK cells [59].

In mouse models, the presence of IL-12 within the melanoma TME has been shown to promote the anti-tumor activity of NKp46^+^ ILC3s against both primary tumors and metastases [60,61] (Figure 1C). Although NKp46^+^ ILC3s acquired an ILC1/NK cell-like phenotype under the influence of IL-12, melanoma rejection was not due to a functional switch, as ILC1s were unable to control tumor growth and key molecules for ILC1 and NK cell anti-tumor activities were dispensable as well. Instead, IL-12-stimulated NKp46^+^ ILC3s induced the expression of adhesion molecules, especially vascular cell adhesion molecule (VCAM), on melanoma vessels, promoting leucocyte influx [60,61]. A similar role has also been proposed for NKp46^-^ ILC3s, which specifically recruited myeloid cells within melanoma [62]. However, the underlying mechanism has not been clarified. Since (i) ILC3s have been shown in vitro to respond to melanoma cells by producing TNFα [23], (ii) melanoma cells express NCR ligands [63] and (iii) TNFα is able to stimulate adhesion molecule expression on endothelial cells [64], ILC3s might likely sense melanoma cells through NCRs and also secrete TNFα in response in vivo, which in turn would promote endothelial activation and leukocyte recruitment.

However, in a CCL21-enriched TME, ILC3s have been proposed to contribute to tumor growth through the generation of a lymphoid-like stroma, which in turn would contribute to establish a tolerogenic environment [65].

## 3. Helper ILC Modulation by Immune Checkpoints Inhibitors in Melanoma

ICIs targeting CTLA-4 and PD-1 have revolutionized the treatment of melanoma and more generally, solid cancers. Although mainly focused on T cell biology, studies in this field indicated that inhibitory receptors are not restricted to T cells and that additional checkpoints are also found in other cells of a lymphoid origin, such as NK cells, for which modulation by ICI therapy and the contribution to its efficacy have been reviewed elsewhere [22]. In this section, we summarize recent findings on the expression of these inhibitory receptors on helper ILCs, the functional modulations triggered by their engagement and the effects of ICIs on ILC subpopulations (Table 1).

### 3.1. CTLA-4

CTLA-4 was the first inhibitory receptor targeted in a clinical setting [11]. In T cells, CTLA-4 is expressed within lymph nodes, where it competes with CD28 and binds CD80/CD86 with a 100-fold higher affinity, thereby turning off the activation [75]. Due to its expression at a central level, inhibitory role in the early phases of activation and ability to modulate both effector and regulatory T cells, CTLA-4 is considered a negative master regulator of the T cell immune response [76].

Analyses performed on peripheral and tumor-resident ILCs from patients with solid cancers, including melanoma, detected variable levels of CTLA-4 expression on all ILC subsets, with tissue cells showing higher expression compared to that in the blood counterpart [24,40]. This suggests that, similarly to that in T cells, CTLA-4 expression may be up-regulated by signals delivered by the TME as an immune-escape mechanism. Consistent with this hypothesis, NK-like ILC1s transdifferentiating from NK cells in murine melanoma showed increased levels of CTLA-4 [42]. In addition, the so-called regulatory ILC2s, expressing IL-10, displayed higher expression of CTLA-4 than their IL-10^-^ counterpart [66]. Therefore, in ILC CTLA-4 might represent a marker of exhausted and/or regulatory states, as has been observed for T cells.

Consistent with the expression of CTLA-4, ILCs also appear to be somehow sensitive to its blockade. Treatment with ipilimumab in melanoma patients has been shown to decrease the abundance of ILC1s and ILC2s in the periphery, whereas the expression of CTLA-4 was reduced in ILC3s and upregulated in ILC1s. Furthermore, patients in whom melanoma disease could not be controlled had a higher percentage of ILC1s than patients in whom disease could be controlled, and therapy was able to significantly decrease this percentage [40]. Overall, these findings support the notion that a peripheral increase in ILC1s might be a pathological event in melanoma and that ipilimumab might restore their numbers. However, this would not be sufficient to confer disease control, pointing out that additional therapies are needed for these patients.

### 3.2. PD-1

PD-1 and its pathway are currently the main target of ICI therapy [77]. In contrast to CTLA-4, PD-1 is expressed on T cells during the effector response; thus it limits T cell activation at peripheral level. PD-1 recognizes two ligands with distinct expression patterns: PD-L1, broadly expressed on cells, and PD-L2, mainly expressed on antigen-presenting cells [78].

In contrast to CTLA-4, PD-1 expression has been detected on mature peripheral and tissue ILCs, as well as on precursors, although it seems to be dispensable for ILC development [24,25,26,27,50,67,68,69,70,71,72,73,74,79]. Despite this broad distribution pattern, the functional role of PD-1 has been mainly investigated in ILC2s, whereas it is scarcely characterized in the other ILC subsets. ILC1s transdifferentiating from NK cells have been shown to up-regulate PD-1 [42]; however, its functional meaning is not known. With regard to ILC3s, PD-1-expressing cells could be found to be enriched in the pleural effusion of malignant patients [73]. Functionally, PD-1 engagement has been demonstrated to be effective in counteracting NCR-mediated activation and reducing cytokine secretion [73,74].

On ILC2s, PD-1 is expressed upon cytokine-mediated activation, especially after IL-33 challenge. This suggest that, similarly to that in T cells, PD-1 may restrain ILC2 activity in order to limit hyperinflammation and tissue damage. Indeed, ILC2s that expressed PD-1 showed a reduced capability to proliferate and produce cytokines (Figure 2A). Accordingly, ILC2s from PD-1-deficient mice displayed improved survival and effector functions [50,67,68,69,70]. Moreover, in several solid cancers, PD-1 expression on ILC2s has been found to be increased in tumor lesions compared to that in healthy tissues, as well as in advanced stages of the disease, indicating that the TME may promote the expression of PD-1 [25,70,71,72].

The effects of the expression and blockage of PD-1 on ILC2s in melanoma have recently been studied in both mice and humans. In mouse models, PD-1 deficiency on ILC2s promoted tumor control by enhancing the intra-tumoral recruitment and secretory capability of ILC2s both in primary and metastatic lesions; similar findings were obtained when mice were treated with PD-1-blocking antibodies [50,53] (Figure 2B). Accordingly, nivolumab therapy in metastatic melanoma patients induced an increase in peripheral mature CD117^-^ ILC2s and increased the ability of ILC2s and ILC3s to produce IL-13 and TNFα, respectively [23]. Moreover, after nivolumab therapy, a low frequency of this subset in the periphery was positively correlated with improved survival in these patients [24]. Collectively, these findings indicate that PD-1 blocking is able in inducing ILC2 expansion and cytokine secretion but the clinical benefit would rely on ILC2 capability to effectively migrate within the tumor lesions in order to exert their anti-melanoma effects. This in turn suggests that combining the anti-PD-1 ICIs currently used in the clinical setting with additional pharmacological agents that promote ILC2 infiltration into the tumor may be a valuable therapeutic strategy for those patients that do not respond to the therapy.

## 4. Helper ILC Modulation by Targeted Therapy in the Melanoma Microenvironment

Mutations within the MAPK pathway, particularly at the BRAF level, are classically known to cause aberrant activation of signaling, independent from growth factors, which causes the uncontrolled survival and proliferation of melanoma cells [3]. However, subsequent research has shown that the effects of such dysregulation are not limited to growth but also contribute to producing a “cold”, suppressive TME [18]. Targeted therapy has been shown to restore a competent microenvironment by inhibiting the production of suppressive cytokines, such as IL-6 and IL-10, and by reducing the infiltration of MDSCs and Tregs [80,81,82].

As for ICIs, ILC2s might be the main helper ILC subset benefitting from the immunological effects of targeted therapy (Figure 3). MDSCs, especially those of polymorphonuclear origin, have been shown to inhibit cytokine secretion by ILC2s via cell–cell contacts [83] (Figure 3A). A similar suppressive effect was exerted by Tregs, although in this case, the inhibition also involved soluble mediators, such as IL-10 and TGF-β [84,85,86] (Figure 3B). Moreover, IL-10 in itself, which is produced by melanoma cells as an autocrine factor [87], appeared to be a more potent inhibitor of ILC2 secretory activity than TGF-β [88,89] and is known for its capability to further stimulate the suppressive activity of both Treg and MDSCs [23] (Figure 3C). Within the TME of mutated melanomas, all these elements would act in concert to suppress the secretory activity of ILC2s and in turn their capability to recruit eosinophils needed for the anti-tumor response [50,51] (Figure 3D). Therefore, by restoring a proficient TME, targeted therapy might indirectly stimulate the anti-melanoma function of ILC2s.

On the other hand, mutated melanomas are enriched in IL-6, which has been shown to promote IL-17 secretion by ILC3s [90]. IL-17 is a pro-inflammatory cytokine involved in many inflammatory and autoimmune diseases. In melanoma, IL-17 up-regulation has been associated with proliferation, invasiveness and angiogenesis [91]. Thus, by dampening IL-6 production by melanoma cells, targeted therapy might counteract the pro-tumoral properties of ILC3s and skew them towards an anti-melanoma phenotype.

Of notice, both primitive and acquired resistance to targeted therapy have been associated with an increased capability of melanoma cells to produce both IL-6 and IL-10 [92,93,94]. Additionally, IL-6 has been demonstrated to further boost IL-10 secretion [95]. This suggests that targeting these two cytokines could be a valuable strategy to overcome resistance and preserve the sensitivity to targeted therapy in metastatic melanoma patients. Indeed, because of the role of IL-6 and IL-10 in promoting the survival and metastasization of cancer cells, as well as in generating a suppressive TME, blocking antibodies directed against them have been developed and are currently under evaluation in different tumor settings, showing promising results [96,97].

## 5. Role of Helper ILCs in Cutaneous Adverse Effects Occurring during ICI Therapy

Reactivation of the immune response triggered by ICIs is associated with the onset of immune-mediated collateral effects, known as immune-related adverse effects (irAEs). Because melanoma is primarily a skin disease, the majority of irAEs affect the skin, likely due to an aberrant autoimmune reaction against antigens expressed by melanocytes and keratinocytes [98]. The frequencies of cutaneous irAEs in metastatic melanoma patients treated with ipilimumab or anti-PD-1 antibodies are comparable, although anti-PD-1 therapy usually elicits milder reactions, and they have been associated with improved clinical responses and survival [99]. The most common cutaneous irAEs are vitiligo and rash, a general term indicating various conditions, such as eczematous patches and psoriasis-like lesions [100]. Although helper ILC presence and/or functions in cutaneous irAEs induced by ICIs have not been directly assessed, the similarities between such manifestations and other clinical entities are suggestive of an active role of helper ILCs in their pathophysiology.

Eczematous lesions resemble those observed in atopic dermatitis (AD), a Th2-mediated skin disorder largely involving ILC2s. Indeed, ILC2 enrichment and activated phenotypes were observed within skin lesions from AD patients. The same results were observed in mouse models of calcipotriol-induced AD [101,102,103]. The adoptive transfer of activated ILC2s, as well as their in vivo expansion and activation, was able to induce AD-like lesions without additional external stimuli [104,105]. Conversely, the genetic ablation of ILC2s or treatment with depleting antibodies was able to relieve the inflammation [101,102]. IL-33 has been shown to be the most potent cytokine activating human skin ILC2s [101], while in mouse models, IL-5 has been demonstrated to be the main mediator recruiting eosinophils to lesions [104,105]. Thus, similarly to what occurs in melanoma [51], ILC2 targeted by anti-PD-1 antibodies might also become hyper-activated and recruit eosinophils in the skin, contributing to the onset of the eczematous AEs observed in ICI-treated melanoma patients.

On the other hand, ILC3s are involved in the pathogenesis of psoriasis. Psoriatic lesions have been found to be enriched in NCR^+^ ILC3s, which correlates with disease severity [106,107,108]. Accordingly, in a humanized mouse model of psoriasis, the transfer of activated NCR^+^ ILC3s was sufficient to induce psoriatic lesions [109]. Of note, the observed NCR^+^ expansion in psoriasis appeared to be at the expense of skin ILC2s, which were reduced within the lesions [107], possibly due to environmental factors that promote the transdifferentiation of ILC2s into ILC3s [46]. Since PD-1 is mainly expressed by NCR^+^ ILC3s and inhibits cytokine secretion [73,74], it is possible that in ICI-treated melanoma patients, PD-1 blocking might unleash the inflammatory potential of ILC3s, which in turn contributes to the development of the psoriasis-like lesions observed in patients.

While skin rashes are not specific irAEs, vitiligo-like depigmentation is observed in melanoma patients, and it is likely the result of an immune response directed not only towards melanoma cells but also against normal melanocytes [99]. Indeed, vitiligo lesions usually exhibit the strong infiltration of immune cells, including adaptive T cells, as well as innate NK cells and DCs [110]. However, a role for helper ILCs has been suggested only for ILC1s. Tulic et al. reported an increased presence of ILC1s endowed with an enhanced capability to secrete IFN-γ within vitiligo lesions compared to that in the skin from healthy subjects [111]. In this work, the authors demonstrated that IFN-γ produced by both NK cells and ILC1s was capable of inducing an autocrine chemokine-mediated signaling pathway that promoted both melanocyte apoptosis and T cell reactivity. However, the specific effect of ILC1s against melanoma cells was not tested. Since it is not known whether and how ICIs can affect the ILC1 phenotype and function, a putative general mechanism of reactivation can only be speculated.

## 6. Role of Helper ILCs in Novel Therapeutic Approaches for Metastatic Melanoma

In recent years, innovative strategies have been proposed for the treatment of metastatic melanoma. These include a combination of ICIs and targeted therapy, novel ICIs and immune agonists and targeted treatment of the TME [15,16,17,18].

The rationale for combined therapies is to exploit their different mechanisms of action in order to improve patient responses. Since the presence of a suppressive TME is one of the main mechanisms negatively affecting ICI efficacy, the combination with targeted drugs would help to generate a permissive TME and to maximize their effects. Indeed, early-phase clinical trials combining anti-PD-1/PD-L1 monoclonal antibodies with both BRAF and MEK inhibitors showed very promising results in terms of survival [16,17]. As usual, in these trials, the immunological response was assessed in terms of T cell expansion and activation, but an active role of helper ILCs cannot be excluded. By inhibiting IL-10 secretion, as well as the recruitment of Tregs and MDSCs, targeted therapy could abolish the inhibition of ILC2s, while IL-6 down-regulation would prevent ILC3s from secreting deleterious IL-17 [80,81,82]. Then, anti-PD-1 antibodies might further enhance the anti-tumor activities of ILC2s and possibly ILC3s [50,53,67,68,69,70,71,72,73,74].

The efficacy of current ICIs together with the evidence that they can induce the expression of additional inhibitory receptors strongly prompted the development of further ICIs. Among the targets are LAG-3, Tim-3, TIGIT and NKG2A, which are under investigation in early clinical trials when combined with anti-CTLA-4 or anti-PD-1/PD-L1 antibodies. As mentioned above, they were all developed based on the biology of T and NK cells and are indeed expressed to different extents by both subsets [15,16]. However, they are also expressed by helper ILCs, particularly ILC1s [35,40,72,112,113,114]. In particular, NK cell-derived ILC1s showed the upregulation of LAG-3, TIGIT and NKG2A under the influence of TGF-β in the melanoma TME [42,43]. However, it is not known whether their blocking is able to revert the phenotype. In humans, the acquisition of NKG2A has been associated with the transition from ILC3s to ILC1s and NK cells [59,113], but its precise role is controversial. Indeed, NKG2A has been shown to label circulating NK-like ILC1s equipped with cytotoxic capability, as well as NK cells more prone to acquire an ILC1 phenotype [35,44,114]. Since the pathological over-expression of NKG2A impaired ILC1 cytotoxicity [114], it is possible that NKG2A marks an intermediate phenotype between ILC1s and NK cells and similar to the latter, acts as an inhibitory receptor to limit hyper-activation [22,23].

On the other hand, Tim-3 has been shown to be specifically expressed by ILC3s, on which it counteracted activation mediated by NCRs. Of notice, Tim-3 and PD-1 are not in principle co-expressed by ILC3s, suggesting that the two receptors mark distinct ILC3 subtypes with specific properties [74]. This is in contrast with what has been reported for TIGIT on ILC2s, for which expression was closely associated with PD-1 [72].

In contrast to ICIs, agonistic antibodies engage co-stimulatory receptors required for cell activation. Among them, the most well studied is the ICOS:ICOS-L pathway, which strongly enhances the T cell-mediated response. Indeed, ICOS targeting is under investigation in combination with anti-PD-1 therapy in different solid malignancies [15]. ILC2s have been shown to express both ICOS and ICOS-L, and the signaling pathway is essential for IL -33-mediated survival, proliferation, cytokine secretion and eosinophil recruitment [115,116]. Thus, ICOS agonists might synergize with anti-PD-1 antibodies in order to improve the anti-melanoma properties of ILC2s.

Another co-stimulatory receptor under investigation in combination with current ICIs is the glucocorticoid-induced TNF receptor (GITR), of which engagement inhibits Tregs and promotes, at the same time, effector T cell functions [15,16]. Interestingly, a similar dual role has been also reported for helper ILCs. Indeed, IL-33-activated ILC2s have been demonstrated to express high levels of GITR, of which engagement by agonistic antibodies further enhanced their capability to secrete the cytokines needed for eosinophil recruitment, such as IL-5 and GM-CSF [117]. On the other hand, GITR appeared to act as an inhibitory receptor on ILC1s, dampening cytokine production [118]. Collectively, this evidence suggests that agonistic antibodies targeting GITR may have a dual beneficial role towards helper ILCs in melanoma, by promoting the anti-tumor activities of ILC2s while counteracting the pro-tumoral capabilities of ILC1s.

Among the drugs that modulate the tumor TME, adenosine inhibitors are those most likely to affect the biology of helper ILCs. Adenosine is a key molecule generating a suppressive TME in melanoma, which is produced by the ectoenzymes CD39 and CD73 and acting by bending several receptors [23]. Accordingly, inhibitory antibodies targeting both the ectoenzymes and the adenosine receptors are currently being evaluated for combinatorial regimens in association with anti-PD-1/PD-L1 antibodies [15]. Adenosine has been shown to negatively affect both ILC1s and ILC2s. In vitro experiments demonstrated that adenosine produced by melanoma could inhibit IFN-γ production by ILC1s, while TNFα secretion was poorly affected [34]. On the other hand, in a mouse model, ILC2s were shown to produce adenosine, which impaired the cytotoxic activity of NK cells [54]. Similar to GITR agonists, drugs targeting adenosine signaling may have a positive dual function on helper ILCs in the context of melanoma, restoring the anti-tumoral properties of ILC1s and counteracting the potential deleterious effects of ILC2s.

## 7. Conclusions

The development of ICIs capable of restoring the T cell response and targeted therapies that specifically target signaling pathways abnormally activated in melanoma cells has revolutionized the clinical approach to the treatment of metastatic melanoma. Subsequent studies demonstrated that these inhibitory receptors targeted by immunotherapy are not limited to T cells but are also present in other cells of innate immunity, such as NK cells and helper ILCs. While the expression patterns of CTLA-4 and PD-1 on NK cells and the role of this subset in melanoma immunotherapy are well known [22], similar knowledge for helper ILCs is still in its infancy. Similarly, the ability of targeted therapy to modulate immune functions is well characterized for T and, to a lesser extent, NK cells [18], but evidence is lacking about the other subsets of ILCs.

The identification of helper ILCs as immune cells involved early in tissue defense and homeostasis has sparked interest in the potential role of these populations in tumor development and progression. Based on the effects exerted under homeostatic conditions, ILC1s have been considered anti-tumor cells, while ILC2s and ILC3s were supposed to be deleterious [28]. However, current results are contradictory and show that the same ILC subset can be both pro- or anti-tumoral depending on the specific tumor. In particular, in melanoma, the activities of helper ILCs appear to diverge from the canonical view, with ILC1s exerting a pro-tumoral effect, whereas ILC2s and ILC3s would be mainly protective. This difficulty in assigning a clear role to helper ILCs is due to different factors. Helper ILCs lack reliable markers that allow identification and manipulation in mouse models and human patients. In addition, ILCs have been shown to be highly plastic and to easily switch from one phenotype and function to another depending on the stimuli they receive in the TME [20]. Therefore, defining markers to accurately recognize the different ILC subsets, developing tools for their manipulation in animal models and identifying the functional molecules produced by helper ILCs in the context of melanoma are essential to characterize their contribution to the TME and in turn, to disease progression.

At the functional level, the role of immune checkpoints on helper ILCs and the effect of their blockage is still poorly defined. Both CTLA-4 and PD-1 have been found to be expressed by all subsets, but only the function of PD-1 and related blocking on ILC2s have been characterized in mouse models, while there are no data on human samples. In ILC2s, PD-1 engagement appears to recapitulate the effects observed on T cells; thus, its blocking promotes melanoma rejection, but the effect on intra-tumoral ILC1s and ILC3s has not been investigated. On the other hand, CTLA-4 has been evaluated only in terms of expression but not function, either with in vitro or mouse models. Therefore, additional experiments are needed in order to clarify the molecular mechanisms modulating the expression of PD-1 and CTLA-4 on helper ILCs, as well as the effects of their blocking by ICIs on the ILC phenotype and function.

Regarding targeted therapy, its effects on helper ILCs has not been directly investigated and can be only speculated based on the previous knowledge of ILC biology. However, since helper ILCs share several similarities with their adaptive counterpart, the restoration of a proficient TME would similarly improve their function [18,20].

In our opinion, four fundamental tasks need to be addressed in order to understand helper ILC biology in a therapeutic setting: (i) the precise characterization of helper ILC subsets in terms of phenotype and function; (ii) development of tools for their manipulation in animal models, as well as human settings; (iii) identification of the molecular pathways regulating immune checkpoint expression; (iv) characterization of the effects of immune checkpoint engagement and blocking on helper ILC functions.

Overall, the gained knowledge would be helpful not only to clarify the role of helper ILCs in melanoma therapy, which is probably underrated, but also and more importantly, to identify additional molecular pathways to be targeted. Indeed, many other immunomodulatory agents, such as antagonistic and agonistic antibodies, which are currently under evaluation for metastatic melanoma therapy, target molecules expressed by helper ILCs and/or mediating their functions. Moreover, helper ILCs express inhibitory receptors that could represent additional targets to be tested in a clinical setting [119].

Combined with current ICIs and/or targeted therapies, drugs modulating ILC plasticity toward an anti-tumor phenotype and/or targeting their effector molecules might thus represent an innovative therapeutic strategy to improve the outcome of metastatic melanoma patients. This also implies that the precise quantification of the different helper ILC subsets in the periphery and/or tumor lesions of metastatic melanoma patients might act as biomarker to predict patient outcomes.

## Figures and Tables

**Figure 1 cancers-15-00933-f001:**
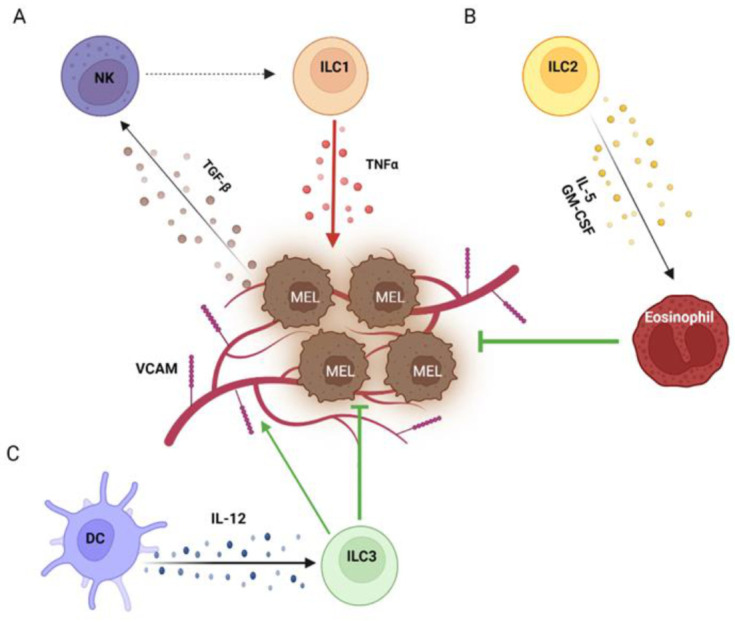
Helper ILCs in the melanoma TME. Within the melanoma TME, TGF-β secreted by melanoma cells switches NK cells into ILC1s, which in turn support melanoma growth by producing TNFα (**A**). On the other side, ILC2s exert anti-melanoma effects by recruiting eosinophils through IL-5 and GM-CSF (**B**). Moreover, ILC3s activated by DCs via IL-12 induce the expression of VCAM on the melanoma vasculature, promoting leukocyte infiltration (**C**).

**Figure 2 cancers-15-00933-f002:**
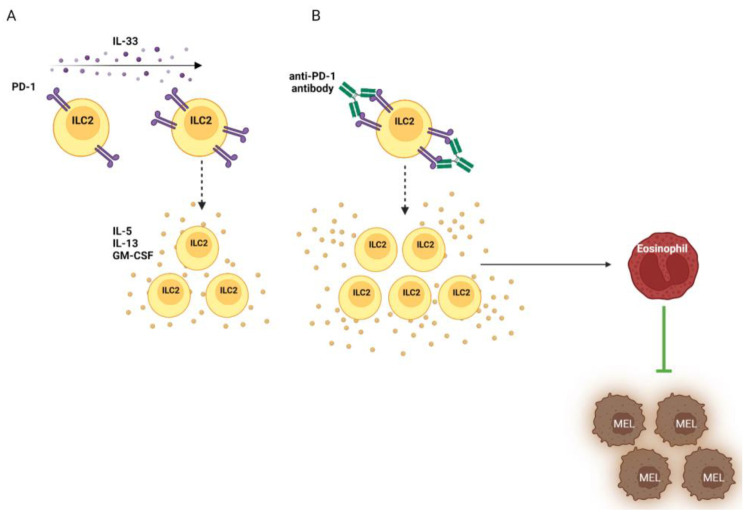
Effects of PD-1 expression and blocking on ILC2s. IL-33-activated ILC2s up-regulate PD-1 in order to limit proliferation and cytokine secretion (**A**). In the melanoma TME, PD-1 blocking by monoclonal antibodies counteracts such restraint, allowing ILC2s to proliferate and produce the cytokines promoting eosinophils infiltration (**B**).

**Figure 3 cancers-15-00933-f003:**
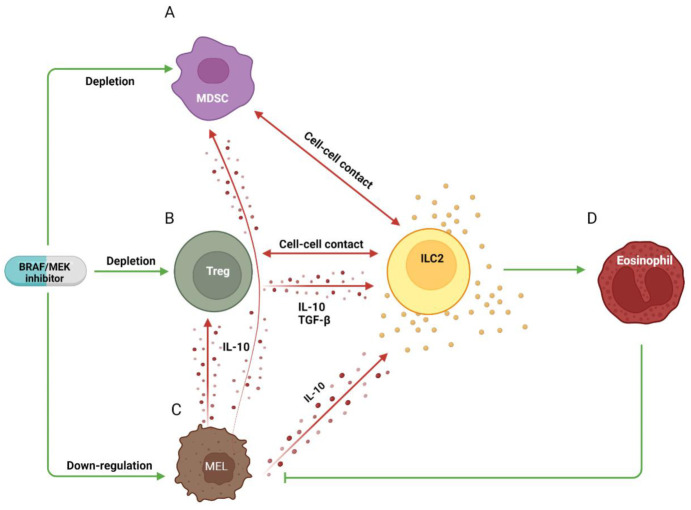
Immunomodulatory properties of BRAF/MEK inhibitors affecting ILC2s. BRAF/MEK inhibitors limit tumor infiltration of MDSCs and Tregs, which may dampen ILC activation through cell–cell contact and suppressive cytokines, such as TGF-β and IL-10 (**A, B**). Moreover, targeted therapy directly down-regulates IL-10 secretion by melanoma cells (**C**). Altogether, these phenomena would contribute to the ILC2-mediated infiltration of anti-melanoma eosinophils (**D**).

**Table 1 cancers-15-00933-t001:** Helper ILCs in melanoma. The table summarizes the current knowledge about the role of the different helper ILC subpopulations in melanoma and how they are affected by ICIs.

Helper ILC Subset	Role in Melanoma	CTLA-4 Function	ICI Effect on CTLA-4	PD-1 Function	ICI Effect on CTLA-4	References
ILC1	Pro-tumoral	Putative exhaustion marker	Putative reversal of activation	Unknown	Untested	[23,24,25,26,34,40,42,43]
ILC2	Anti-tumoral	Putative exhaustion marker	Putative reversal of activation	Inhibition of activation	Reactivation of proliferation and secretory activity	[23,24,25,26,40,49,50,51,52,53,66,67,68,69,70,71,72]
ILC3	Anti-tumoral	Putative exhaustion marker	Putative reversal of activation	Inhibition of activation	Putative reactivation	[23,24,25,26,40,60,61,62,73,74]

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
