# Peer review of "Helper Innate Lymphoid Cells—Unappreciated Players in Melanoma Therapy"

_cancers, 2023, doi:10.3390/cancers15030933_

Round 1

Reviewer 1 Report

This is a very nice review focusing on the role of innate lymphoid cells in melanoma. The review is supported by well-designed figures. The authors also discussed the influence of currently available treatments on innate immune cells in melanoma environment. Maybe the authors can also discuss shortly the new anti-melanoma treatments that are currently evaluated , but not yet registered in terms of analysed topic. 

Author Response

We thank the reviewer for the very positive evaluation of our work. According to his/her suggestion, we added a paragraph (now Paragraph 6, pages 11-12, lines 467-536) describing innovative therapeutic strategies for metastatic melanoma that are currently under investigation and may involve processes mediated by helper ILCs

Reviewer 2 Report

Excellent review on ILCs, very thorough and logical. Figures quite helpful. Literature adequately reviewed. Two topics not covered enough:

1 - Role of ILCs in skin manifestations of melanoma patients treated with ICI

2 - Role of ILCs in melanoma-induced vitiligo

Author Response

We thank the reviewer for the very positive evaluation of our work. According to his/her suggestion, we added a paragraph (now Paragraph 5, pages 10-11, lines 417-465) describing the possible role of ILC in cutaneous events occurring during ICI therapy

Reviewer 3 Report

The review article detailly described and justified the title helper innate lymphoid cells_unappreciated players in melanoma-therapy, however, some points could be considered for a better understanding to the readers making it more reader-friendly.

1)The comparison and advantage of this review to another similar review article should be kept or discussed in a few lines to highlight the key points or novelty of this review article in the last paragraph of the introduction.

2) Comparison of ILC1/ILC2/ILC3 in melanoma and ICIs CTLA-4 and PD-1 should be tabulated briefly with key points highlighting the differences.

3)Conceptual future insights or outlooks from the author's point of view should be added in the last section of this review article.

Author Response

We thank the reviewer for the positive evaluation of our work and useful comments to improve it.

1) A brief comparison between our work and other reviews in this field is now provided at the end of the Introduction (Page 1 lines 87-93).

2) we have taken the Reviewer’s suggestion and added a table summarizing the key points regarding ILCs and ICIs in melanoma (Table 1, Page 7).

3) as the reviewer has rightly pointed out, we have improved the discussion with our point of view on the future clinical application of ILCs (Pages 13-14, lines 579-597).